# Clinical Aspects and Therapeutic Management of an Aggressive Manifestation of Stage III Grade C Periodontitis in a Female Teenager

**DOI:** 10.3390/diagnostics13061077

**Published:** 2023-03-13

**Authors:** Stana Păunica, Marina-Cristina Giurgiu, Dragoș Nicolae Ciongaru, Cristiana-Elena Pădure, Ștefan Dimitrie Albu, Silviu-Mirel Pițuru, Anca Silvia Dumitriu

**Affiliations:** 1Department of Periodontology, Faculty of Dental Medicine, University of Medicine and Pharmacy “Carol Davila”, 050474 Bucharest, Romania; stana.paunica@umfcd.ro (S.P.); stefan-dimitrie.albu@drd.umfcd.ro (Ș.D.A.); anca.dumitriu@umfcd.ro (A.S.D.); 2Doctoral School, Faculty of Dental Medicine, University of Medicine and Pharmacy “Carol Davila”, 050474 Bucharest, Romania; cristiana-elena.padure@drd.umfcd.ro; 3Department of Professional Organization and Medical Legislation-Malpractice, Faculty of Dental Medicine, University of Medicine and Pharmacy “Carol Davila”, 050474 Bucharest, Romania; silviu.pituru@umfcd.ro

**Keywords:** antibiotics, bone loss, gingival overgrowth, molar/incisor pattern, periodontal therapy, aggressive periodontitis, periodontitis management, periodontitis in teenager

## Abstract

The main objective of this study was to evaluate the improvement of periodontal health in patients with periodontitis treated with non-surgical periodontal therapy and subgingival-administrated local and systemic antimicrobial agents. A female teenager with periodontitis-associated health issues and a history of dental trauma was selected for this study. Clinical indices were obtained, and radiographic examination was performed at the beginning of the study. The patient was treated with periodontal therapy and administration of antibiotics. After this therapy, visits were scheduled at regular intervals to observe the clinical changes. Non-surgical periodontal therapy and administration of local and systemic antibiotics resulted in a reduction in the patient pocket depth probing, plaque index, and bleeding on probing. Gingival and periodontal health improved in terms of gingival overgrowth, plaque, tartar index, and tooth mobility. Suppuration was eliminated, and no gingival inflammation signs were observed.

## 1. Introduction

An understanding of the etiology and pathogenesis of periodontitis is essential for treatment planning. An attempt to classify periodontitis was presented in 1999 by the American Academy of Periodontology committee on the classification of periodontal diseases. They concluded that periodontal diseases could be classified as chronic or rapidly progressing diseases [1,2,3,4,5]. The new classification from 2018 grouped the periodontal pathology according to the stages and degrees of progression so that the aggressive forms from the 1999 Armitage classification [2] were assimilated into the advanced degrees of progression. In the case presented, we encountered this kind of periodontal pathology in a 15-year-old young patient. This new classification allows an early and rapid diagnosis that will offer better clinical outcomes.

In the recent years, the concept of “full-mouth disinfection” within a 24 h period and the association of adjunctive antibiotics and antiseptics with scaling and subgingival debridement to improve the clinical outcomes were introduced [5,6,7].

Successful treatment of localized and generalized periodontitis has been shown to depend on the reduction of specific pathogenic bacteria [8,9].

Traumatic dental injuries are a significant challenge, since they often result in early damage of the teeth and their supporting tissues, frequently leading to an unfavorable prognosis that may lead to tooth loss [10]. Dental trauma is particularly challenging in children, whose tooth development and jaw bone growth are incomplete and where low compliance may influence the effectiveness of the appropriate treatments [11].

Additionally, at adolescence, the hormonal change can have an influence on the development of subgingival microbiota with gingival clinical outcomes. Pubertal gingivitis is characterized by the inflammation of the gums, increased gingival volume due to edema, and bleeding when chewing and brushing. Increased gingival volume is common in the labial area at the level of the interdental papilla. After puberty, gingival inflammation and overgrowth tend to decrease.

## 2. Materials and Methods

### 2.1. Case Presentation

A 15-year-old female teenager was referred by a general practitioner to the Department of Periodontology, Faculty of Dental Medicine, University of Medicine and Pharmacy Carol Davila, Bucharest, Romania. There were no other diseases, such as diabetes or other hematological disease, no smoking, and no medications being taken by the patient. She had an accident (falling down the stairs) when she was 7 years old but did not experience any general or dental trauma (without notable objective or subjective clinical signs) as a result. Consequently, she did not visit a dentist for any treatment. The reasons for the first visit were teeth mobility, which appeared 2 years behind, poor aesthetic aspect of gingiva, and poor masticatory performance. The following clinical parameters were assessed at baseline: bleeding upon probing (BOP), plaque index score (O’Leary et al.) [12] and probing depth (PD). All measurements were performed by the same examiner using a periodontal probe (North Carolina 15 mm probe) on six sites for each tooth. The initial examination revealed deep probing pocket depth, severe gingival overgrowth, bleeding upon probing, and gum recession. Gingival inflammation was observed. Periodontal abscesses were registered to both maxillary central incisors (Figure 1). Suppuration and tooth mobility were registered to both maxillary central and lateral incisors. Gingival overgrowth was registered mainly on the maxillary anterior tooth side.

The plaque index score registered and bleeding upon probing (BOP) were 100%. The initial probing depth (PD) was 10 mm at the left central incisors (mesial) and 8–9 mm at the other proximal sites of the both upper central incisors. The tooth mobility (Miller Classification) was Class 2 for both the maxillary central and lateral incisors (Figure 2).

The initial radiography showed severe bone loss on the maxillary anterior tooth (Figure 3). In addition to the periodontal aspects, dental malposition of the maxillary and mandibular central and lateral incisors, deep bite malocclusion, deep hard palate, and advanced carious lesion in both maxillary first molars were observed. It was also not possible to specify whether the bone resorption in the anterior area was related to the trauma, considering that after the trauma, the patient did not request dental treatment.

The periodontal diagnostic was stage III, grade C periodontitis. Stage III of periodontitis, in this case, was defined based on the severity (interdental CAL ≥ 5mm, radiographic bone loses extending to the apical third of the root) and complexity of management (probing depth > 6 mm, vertical and horizontal bone loss, tooth mobility degree ≥ 2, and masticatory dysfunction). The grade of periodontitis was estimated with direct or indirect evidence of % bone loss/age > 1.0 and destruction exceeding biofilm deposits (suggestive of rapid progression and early onset disease) [12,13,14]. The orthodontic diagnostic was class II, division 1 malocclusion (Angle classification) and deep bite malocclusion, with dental malposition primary and secondary to the periodontal disease. The smile line was altered due to the overgrowth of gingival tissue at the maxillary tooth. Additionally, following vitality and percussion tests, both the maxillary and mandibular incisors were vital and without any signs of pulpal involvement.

The dental diagnosis was simple and complicated multiple carious lesions, which involved the first molars. The occlusal diagnosis was protrusive working interferences at the level of both maxillary central incisors, due to their accelerated eruption secondary to periodontal disease.

### 2.2. Treatment

The patient accepted the proposed periodontal treatment plan and signed the informed consent form.

The periodontal treatment started with the drainage of periodontal abscesses and lavaging with antiseptic substances: chloramine solution 3‰ (3 per-mille) and hydrogen peroxide solution 3% [15]. The treatment plan continued with the antimicrobial phase after the initial examination and consisted of several sessions. Initial treatment began with the removal of supragingival plaque using small gauze on the tooth and the gingival surface associated with antiseptics (chloramine solution 3‰ and hydrogen peroxide solution 3%) and local antibiotics (prescription prepared at the pharmacy and consisted of Tetracycline 3 g, Metronidazole 3 g, and Glycerin as a vehicle). This product was administered with a syringe into the periodontal pocket under isolation. It was administered four times at three-day intervals [16,17]. Following this step, the subgingival debridement was performed. The quadrant subgingival debridement was performed by the same examiner using hand instruments (Gracey curettes) 

After the first therapy session, the patient was instructed on how to improve his oral hygiene and recommended to use, at home, an adjuvant topical antimicrobial product, which was also prepared at the pharmacy [16], twice a day for 10 days and a systemic antibiotic Augmentin 1 g (amoxicillin and clavulanic acid) twice a day for 7 days. Additionally, this product was recommended for use after oral hygiene and applied at home with a small piece of gauze in 8-10 circular motions from the base to the tip of the papilla and in linear motions along the gingival margin. This topical antimicrobial product was a mixture of two antibiotics (neomycin and metronidazole), hydrocortisone acetate, stamycin, and vitamin A, with glycerol as the vehicle [16]. The next therapy session was scheduled after 3 days, and what was performed in the first phase was repeated (Figure 4).

The third and fourth therapy sessions were performed 1 week apart and consisted of scaling and root planing with hand and ultrasonic instruments (SRP), subgingival local administration of antiseptics [15,16], and subgingival local administration of the previous prescription (tetracycline 3 g and metronidazole 3 g, with glycerin as a vehicle). In the third session (Figure 5), it was decided to continue the antibiotic administration with azithromycin 500 mg once a day for 3 days [15]. It was considered necessary to recommend this antimicrobial product to reduce gingival inflammation and suppuration. Tetracycline, on the antibiotic spectrum, is inferior to amoxicillin, but it has anti-collagenolytic properties that inhibit the MMPs (matrix metalloproteinases), has an antioxidant role, and as metronidazole, it can be administered locally with minimal secondary effects [17]. SRPs were performed in multiple sessions, due to gingival overgrowth. With each antimicrobial treatment session, the gingival overgrowth decreased, and SRP was performed more efficiently.

After 3 months of phase 1 of periodontal treatment, the patient returned to the clinic for supportive periodontal therapy (Figure 5). The clinically measured gingiva recession was about 6 mm for the maxillary central incisors and 4 mm for the lateral incisors. According to the Miller classification—class III gingival recession—vestibular marginal tissue recession extends to the mucogingival junction. Loss of interdental bone is apical to the CEJ but coronal to the apical extent of the vestibular marginal tissue recession (Figure 5).

The mobility of the teeth (Miller classification) was maintained at Class 2 for the two maxillary central incisors. This mobility was a result of inflammatory edema but also of the high degree of alveolar bone loss (Figure 6). The patient had no discomfort or pain.

Given the increased tooth mobility, a composite splint was applied to the facial and interproximal surfaces of the central and lateral incisors (Figure 7). The patient received indications related to the improvement of oral hygiene by using additional products (oral irrigators and interdental brushes). The patient was advised to present for follow-up (supportive periodontal therapy) after 3 months in the first year.

After this treatment, the patient was able to improve their dental hygiene, as the teeth were splinted with composite resin. SRP, in combination with the administration of antiseptics and antibiotics in this case, seemed to be an effective approach to improve periodontal health in severe stage III, grade C periodontitis.

## 3. Discussion

In this clinical case presented, several factors favored the onset and progression of periodontal disease: trauma, severe carious lesions that led to the loss of the maxillary first molars, deep bite malocclusion, and severe dental malposition. In choosing a treatment plan, the patient’s age must also be taken into consideration. In the frontal area, there was a high probability that bone destruction was the result of trauma that, coupled with poor hygiene, allowed the development of a severe periodontal inflammation. Additionally, the procedures for the long-term maintenance of the results obtained at the periodontal level by controlling the bacterial plaque are very relevant. Regularly scheduled dental appointments are important at predetermined time intervals to perform gingival debridement associated with local antiseptics.

It is crucial to highlight the role of bacterial plaque in the progression of gingival inflammation. Another thing to consider is the traumatic accident that she had when she was 7 years old. Given that the patient did not seek dental treatment after the accident, we did not have a detailed dental history to determine whether or not any changes occurred on the alveolar bone. Therefore, the main cause of bone loss was the progression of an inflammatory process caused by plaque accumulation in sites with traumatic injury, long-time progressive subclinical signs, and ignored manifestation.

In addition to the mechanical removal of the formed bacterial plaque and dental calculus, which results from the transformation of the plaque and provides a support for the plaque, antimicrobial substances are also necessary to prevent plaque attachment and to destroy bacterial colonies while preferably maintaining the saprophytic flora. Additionally, these antimicrobial products limit the recolonization of the root surfaces for as long as possible after treatment. Thus, often in periodontal therapy, combinations of broad-spectrum antiseptics and antibiotics are necessary. These have the ability to act both on the gingival tissue and at the gingival sulcus [17,18]. Therefore, some mouthwashes contain active ingredients that can help to control the growth of plaque and reduce the risk of gum disease. In addition to these daily hygiene practices, regular dental follow-ups are also important to maintaining good oral health.

Professional mechanical plaque removal can help prevent the onset and progression of periodontal disease and can also help identify the disease in its early stages, when it is the most treatable [18]. Proper oral hygiene and regular professional dental cleanings are essential to preventing the progression of periodontal disease.

Additionally, the mechanisms of the host immune response need to be considered when talking about periodontal disease. Most of the time, the host is in a symbiosis status with the microorganisms involved in periodontal disease. However, abnormal reactions (severe host immune inflammatory response) can sometimes occur and lead to a break in the symbiotic status. This situation often results in the destruction of the tooth-supporting tissues [19,20,21].

Chronic gingivitis is the most common form of periodontal disease in adolescents. In some cases, gingival overgrowth may occur due to hormonal changes. There are physiological or pathological situations when hormonal changes may occur and can be accompanied by changes at the periodontal level. Physiological changes occur during puberty, pregnancy, and menopause. The hormones responsible for these changes are mainly estrogen and progesterone, which can frequently cause gingival hyperplasia. Additionally, destructive forms of periodontitis (localized or generalized) can be observed but with limited clinical outcomes at the gingival level. Longitudinal studies of disease progression in young adults show that individuals with signs of destructive periodontitis tend to deteriorate further at a young age, particularly with the involvement of *Aggregatibacter actinomycetemcomitans* and *Capnocytophaga sputigena* as periodontal pathogens [21,22].

Although SRP is effective at disrupting subgingival deposits and eliminating bacteria, the rapid recolonization of bacteria after SRP means that disease recurrence is almost inevitable. Therefore, the use of antibiotics is advantageous [23]. While most studies have shown a statistically significant benefit from the administration of systemic antibiotics in combination with nonsurgical periodontal therapy, their long-term clinical significance is still a subject of debate [24]. The actual effectiveness of antibiotic therapy in systemic and localized forms has been called into question, due to the risk of developing microbial resistance, as well as the adverse impact of this therapy on the human microbiome. This concern is especially relevant, given the widespread and indiscriminate use of antibiotics for the treatment or prevention of oral infections [25].

However, as more and more bacteria become resistant to antibiotics, traditional treatments for periodontitis may become less effective. This can result in longer and more complicated treatment regimens, as well as a higher risk of complications and failures.

Additionally, the development and implementation of new and alternative treatments for periodontitis, such as those that focus on reducing the bacterial load or stimulating the body’s own immune response, may help to reduce our reliance on antibiotics and slow the development of resistance [26,27,28,29].

In this current case, due to the polymicrobial etiology of periodontal disease, with a high virulence manifested by inflammation and suppuration collections, the usage of combined antimicrobial products was considered to be necessary. This case report demonstrates how adjunctive antiseptic and antibiotic therapy influences periodontal health. The reasons for changing the class of the systemically administered antibiotic were the persistence of the gingival inflammatory signs, to avoid the development of antimicrobial resistance, and to use an antibiotic (azithromycin) that acts mainly and more on the immune response component.

One of the benefits of azithromycin on the treatment of periodontal disease is its convenient dosing schedule. Contrary to some other antibiotics that require multiple doses per day, azithromycin is typically prescribed as a single dose per day for a short period (3–5 days). This can make it easier for patients to adhere to the treatment regimen and achieve a successful outcome. In addition to its antibacterial properties, azithromycin has also been shown to have anti-inflammatory effects, which can be beneficial for treating periodontal disease [30].

Considering the chosen subgingival debridement method (quadrant-wise), it can be said that numerous studies have shown that there are similar clinical outcomes when it comes to the choice of scaling and the SRP treatment method (quadrant-wise or full-mouth), as long as the procedure is performed correctly [31,32,33,34]. Additionally, microbiological studies conducted after evaluating both techniques also supported the same conclusions [35]. For the future, the possibility of prosthetic or implant treatment must be considered to restore the aesthetics of the upper anterior area. Given the lack of gingival tissue, muco-gingival surgery should also be considered. It is important to consider the biological and functional factors, aesthetics, and cost when determining the best treatment.

## 4. Conclusions

In this case report, the efficacy of SRP and the adjunctive use of antibiotics and antiseptics had the main purpose of avoiding tooth extraction. Considering the age of the patient (15 years old), the antibiotic therapy was adjusted in order to limit adverse effects, so local antibiotic administrations were preferred. Additionally, the repeated and scheduled sessions of debridement for maintaining the clinical outcomes were very important.

## Figures and Tables

**Figure 1 diagnostics-13-01077-f001:**
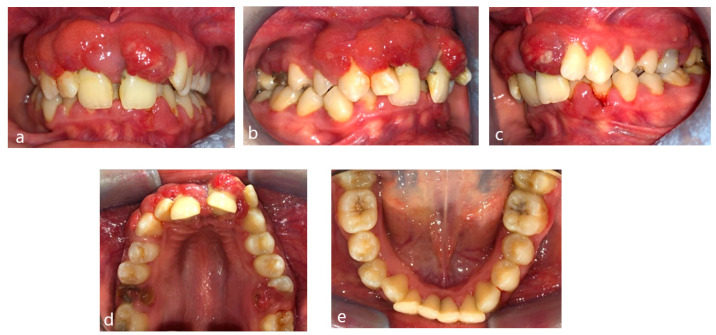
Clinical aspects before periodontal treatment: (**a**) frontal aspect, (**b**) right lateral aspect, (**c**) left lateral aspect, (**d**) maxillary occlusal aspect, and (**e**) mandibular occlusal aspect.

**Figure 2 diagnostics-13-01077-f002:**
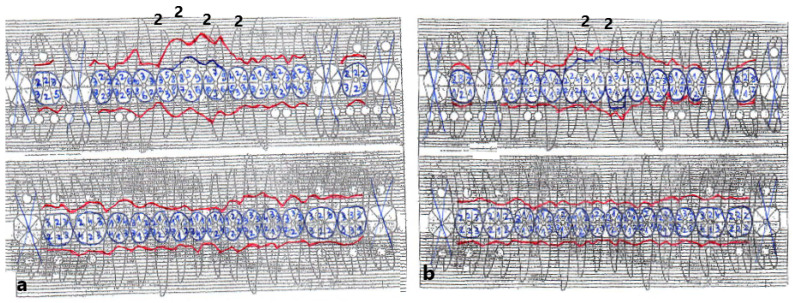
Periodontal chart before treatment (**a**) and after treatment (**b**) (Department of Periodontology, Carol Davila University of Medicine and Pharmacy). Red line—probing depth (PD), blue line—gingival margin, crossed tooth—excluded tooth, class 2—tooth mobility (Miller Classification).

**Figure 3 diagnostics-13-01077-f003:**
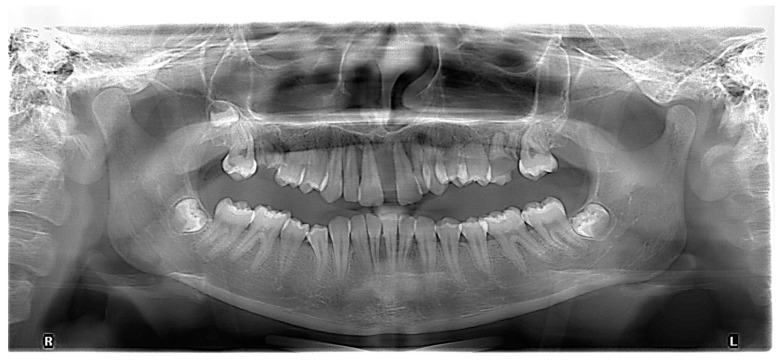
Initial panoramic radiograph.

**Figure 4 diagnostics-13-01077-f004:**
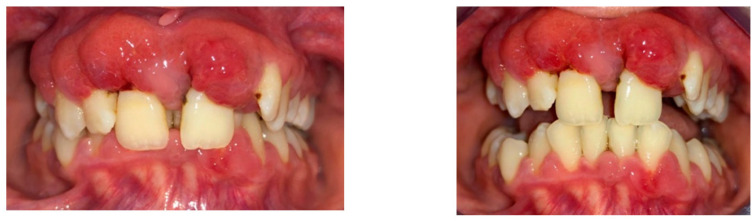
Clinical aspects at the second presentation (3 days after initial examination).

**Figure 5 diagnostics-13-01077-f005:**
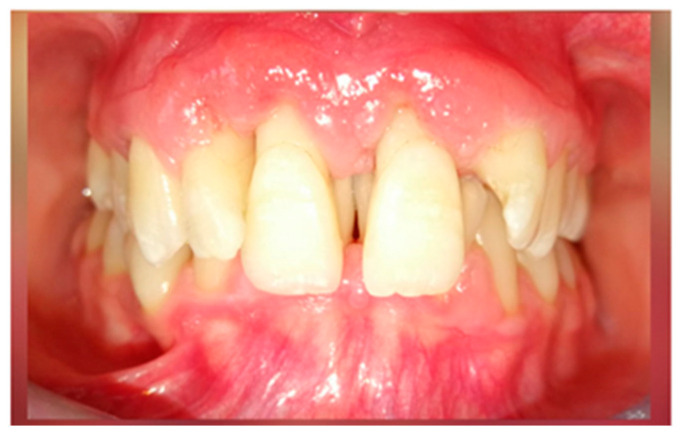
Clinical aspect three weeks after the initial visit and periodontal treatment.

**Figure 6 diagnostics-13-01077-f006:**
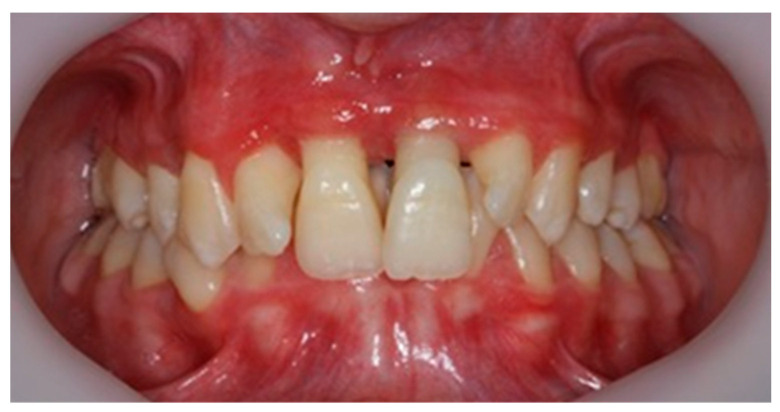
Clinical aspect after 3 months of periodontal therapy.

**Figure 7 diagnostics-13-01077-f007:**
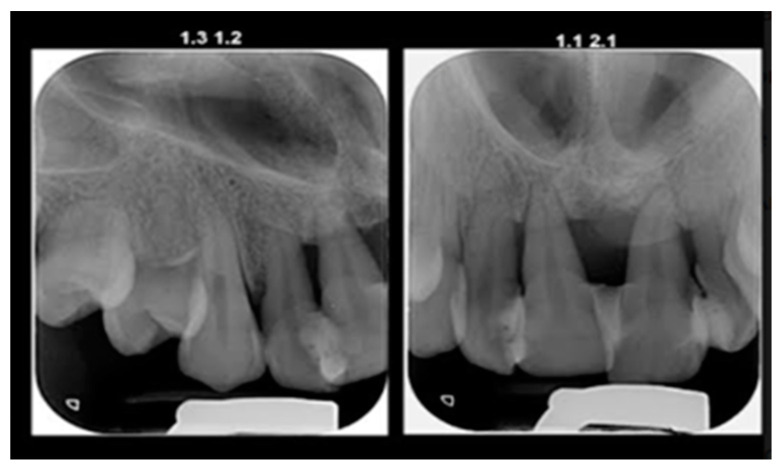
Radiographs after periodontal treatment.

## Data Availability

Data is contained within the article.

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
