# Peer review of "Clinical Aspects and Therapeutic Management of an Aggressive Manifestation of Stage III Grade C Periodontitis in a Female Teenager"

_diagnostics, 2023, doi:10.3390/diagnostics13061077_

Round 1

Reviewer 1 Report

Comments and suggestions are labelled and inserted in the pdf file. 

Author Response

Thank you for your comments and suggestions. Therefore, I have taken into account your recommendations and suggestions, and I have added them to the article. I have reconsidered and modified the manuscript, especially in the discussion and conclusion sections. Furthermore, I have included additional information related to the treatment plan. Please see the new version of the manuscript uploaded to the system. 

Kind regards,

Reviewer 2 Report

Accept after Clarification : 

1: is this bone loss due to periodontitis only or due to a history of trauma or Deep bite?

2: History of the nature of the trauma, any history of the follow-up? 

3: should include an intra-oral x-ray of upper & lower anterior  teeth considering that  (OPG X-RAY) usually not clear in the anterior area 

4: Is there any vitality test done for the Upper and the lower anterior teeth? 

5: Consider including a Full Perio chart.

6: consider adding information for the smile line.

Author Response

Dear Reviewer,

Thank you for your response. It is important for the readers of the article to have as much information as possible. I have taken into account your recommendations and suggestions, and I have added them to the article. Also, I have reconsidered and modified the manuscript, especially in the discussion and conclusion sections. Please see the attachment.

Kind regards,

Reviewer 3 Report

The reviewer really appreciates the efforts of the authors to conduct this study which has good clinical significance. The manuscript is well written without leaving major issues in it. However, The length of the article is bit long for a case report. It is just a suggestion to reduce the length otherwise no other comment

Author Response

Dear Reviewer, 

Thank you for your comments and suggestions. I have taken them into consideration and have reduced the length of the manuscript.

Kind regards,

Round 2

Reviewer 1 Report

I truly appreciate the corrections made by the authors based on my comments in the first round of review. However, the case reported lacks novelty in which it is a common periodontitis case with a common clinical management. Therefore, it is reflected in my overall recommendation.